# Non-Invasive Retinal Blood Vessel Wall Measurements with Polarization-Sensitive Optical Coherence Tomography for Diabetes Assessment: A Quantitative Study

**DOI:** 10.3390/biom13081230

**Published:** 2023-08-08

**Authors:** Hadi Afsharan, Dilusha Silva, Chulmin Joo, Barry Cense

**Affiliations:** 1Optical+Biomedical Engineering Laboratory, Department of Electrical, Electronic and Computer Engineering, The University of Western Australia, Perth, WA 6009, Australia; barry.cense@uwa.edu.au; 2Microelectronics Research Group, Department of Electrical, Electronic and Computer Engineering, The University of Western Australia, Perth, WA 6009, Australia; dilusha.silva@uwa.edu.au; 3Department of Mechanical Engineering, Yonsei University, Seodaemun-gu, Seoul 03722, Republic of Korea; cjoo@yonsei.ac.kr

**Keywords:** PS-OCT, diabetes, blood vessel wall birefringence index (BBI), retinal imaging, retinal blood vessels

## Abstract

Diabetes affects the structure of the blood vessel walls. Since the blood vessel walls are made of birefringent organized tissue, any change or damage to this organization can be evaluated using polarization-sensitive optical coherence tomography (PS-OCT). In this paper, we used PS-OCT along with the blood vessel wall birefringence index (BBI = thickness/birefringence^2^) to non-invasively assess the structural integrity of the human retinal blood vessel walls in patients with diabetes and compared the results to those of healthy subjects. PS-OCT measurements revealed that blood vessel walls of diabetic patients exhibit a much higher birefringence while having the same wall thickness and therefore lower BBI values. Applying BBI to diagnose diabetes demonstrated high accuracy (93%), sensitivity (93%) and specificity (93%). PS-OCT measurements can quantify small changes in the polarization properties of retinal vessel walls associated with diabetes, which provides researchers with a new imaging tool to determine the effects of exercise, medication, and alternative diets on the development of diabetes.

## 1. Introduction

The number of people diagnosed with diabetes has quadrupled during the last 40 years, causing more than two million deaths around the globe in 2019 [1]. Diabetes has a substantial role in blindness, renal and heart failure, stroke and lower limb amputation [2]. Hyperglycemia (high blood glucose) decreases the elasticity of the blood vessels impeding blood flow and supply of oxygen to the organs [3], which in return increases the risk of hypertension and worsens the damage to the vasculature. Since the structure of the blood vessels is altered through molecular mechanisms regulated by diabetic metabolic abnormalities (such as hyperglycemia and insulin resistance) [4], measurements of the extent of these structural changes may help to early diagnose diabetes, to help the patient manage the disease and to prevent microstructural damage and end-organ damage.

Given that the duration of diabetes stands out as a significant risk factor for peripheral vascular complications, retinopathy and neuropathy, diagnosis of diabetes at the earliest stages of the disease is of critical importance [3,5]. Studies show that diagnosis of diabetes at younger ages reduces severity hyperglycemia severity [6,7]. Research suggests that the crucial time for screening for diabetes and corresponding vascular damage is as early as puberty [8]. Furthermore, there is an interlink between cardiovascular disease (CVD) and diabetes, and CVDs remain the major causes of mortality in patients with diabetes [2]. Hence, early diagnosis and controlling of diabetes substantially reduces the risk factors for developing CVD-triggered complications [9,10].

Evaluating the structural and functional characteristics of blood vessels becomes crucial for early detection of vascular disorders [11]. Recent investigations have revealed that alterations in blood vessel architecture can precede the onset of CVDs [12,13]. The loss of blood vessel wall elasticity may arise due to aging, calcification, changes in collagen fibers and elastin composition, or inflammation [14,15]. Clinical studies have indicated that increased blood glucose in patients with diabetes triggers endothelium-dependent vasodilation in blood vessels that may be the underlying cause of atherogenic propensity [16,17]. Plaques developing in the intima layer of the arteries, known as atherogenesis, are usually accompanied by inflammation and fibrosis [18], which exhibits birefringent characteristics [19]. Birefringence is an optical property that reflects the microstructural organization of tissue. Hence, imaging techniques such as polarization-sensitive optical coherence tomography (PS-OCT) which are sensitive to birefringence can be used to study and diagnose diabetes. PS-OCT is a well-established imaging modality capable of detecting fibrosis, in vivo. Gräfe et al. indicated the importance of PS-OCT in subretinal fibrosis in neovascular age-related macular degeneration (AMD) through measurements of birefringent collagen fibers in the retina [19]. In another study, PS-OCT and birefringence measurements were used to assess skin fibrosis in the multisystem disease systemic sclerosis (SSc) and showed that retardation can be potentially used as a suitable biomarker for SSc-related fibrosis [20]. There are also studies that implement PS-OCT for idiopathic pulmonary fibrosis [21,22]. However, no non-invasive studies have been conducted to verify diabetes in vivo through assessing the structural changes in blood vessels and fibrosis. Kuo et al. demonstrated that atherogenesis in affected blood vessels caused increased birefringence when compared to a healthy vascular intima, but their PS-OCT measurements were performed ex vivo [23].

We recently introduced PS-OCT to non-invasively assess the structural integrity of the retinal blood vessel wall in vivo. We quantified the vessel wall’s double-pass phase retardation over unit depth (DPPR/UD), equivalent to birefringence, and the vessel wall thickness, measured close to the optic nerve head (ONH) to identify and distinguish between an artery and vein, as they tend to be paired [24]. We also used this method to investigate the differences between blood vessels of healthy subjects and patients with hypertension [25]. We demonstrated that the blood vessel birefringence index (BBI), a newly introduced index that combines DPPR/UD and thickness of the retinal vessel walls, robustly discriminated between blood vessels of normotensive and hypertensive individuals with high sensitivity and specificity. The goal of the present study was then to implement PS-OCT measurements to compare the polarization properties of retinal blood vessels of patients with diabetes with those of healthy age-matched subjects. PS-OCT measurements were first collected from five diabetic and ten age-matched healthy subjects and analyzed to extract BBI values. These values were then used to identify classifying thresholds that can consequently be used to distinguish healthy blood vessels from blood vessels of diabetic patients.

## 2. Materials and Methods

### 2.1. Experimental Setup and Imaging Protocol

Measurements were performed with a custom-made PS-OCT instrument [24]. Briefly, a fiber-based PS-OCT system was coupled with a home-made spectrometer operating at a center wavelength of 840 nm. The axial resolution of 6 μm was measured on a mirror in a model eye [26]. The lateral resolution of the system was estimated at 12.9 μm owing to the implementation of a 2.4 mm beam diameter (1/e^2^). Defocus and astigmatism were corrected in the sample arm with a Badal optometer and with trial lenses, respectively. The system recorded 4.5 mm by 4.5 mm images, covering the largest blood vessel inferior and superior to the ONH in approximately 4 s. At least two volumetric scans were collected from each eye of the subjects measuring 100 B-scans × 1000 A-scans × 512 pixels in depth.

A pupil camera was also implemented in the sample arm to help center the imaging beam onto the cornea. Our method does not require pupil dilation. Participants were asked to sit in front of the system where a chin- and headrest were provided for the patient’s comfort and improved imaging stability. A fixation target was mounted on a translational stage, to change a subject’s fixation such that the captured images covered the largest blood vessels near the ONH.

### 2.2. Subject Recruitment and Study Approval

Subjects recruited for the study were classified into two cohorts: patients and age-matched healthy individuals. Patients with type II diabetes were identified by their general practitioner and were reported as being on insulin, metformin, or both, meaning that diabetes in our patient’s population was under control (Table 1). Based on the fundus images, none of the recruited diabetic subjects had diabetic retinopathy. None of the participants was diagnosed with underlying systemic problems such as hypertension, heart disease, dyslipidemia or stroke. The patients’ blood glucose level was also checked prior to PS-OCT measurements. Smoking status was also recorded in patients and classified as never, former, or current. The cohort of healthy age-matched subjects was self-identified as healthy; healthy subjects had no history of high blood glucose. PS-OCT imaging measurements adhered to the tenets of the Declaration of Helsinki and were approved by the Human Ethics committee of the University Western Australia (reference number: 2019/RA/4/20/4955). Prior to PS-OCT imaging, informed consent was obtained from all participants.

### 2.3. Retardation and Image Analysis

Retardation images were generated based on a method described previously [24], implementing a Stokes-vector-based analysis. After *k*-space linearization and dispersion compensation, Stokes parameters were calculated [27,28]. Then, an intensity-based threshold was applied to find the surface of the retina [29]. This surface was used as a reference for a relative retardation measurement [30]. During the averaging we made sure that our retardation data were not contaminated by pixels in the vitreous, as these pixels were set to NaN. Next, to find the retardation value of each pixel in depth *z*, Stokes vectors of a pixel were compared to those at the retinal surface. Furthermore, to ensure the retardation analysis was independent of the incident polarization state, the retardation values were angle and intensity weighted [31,32]. Figure 1 shows the resulted cross-sectional retardation images of an area near the ONH of a diabetic patient.

In the next step of the analysis, the polarization properties of the blood vessels were determined with a previously described procedure [24]. This procedure includes several steps to extract DPPR/UD and the thickness of the blood vessel, shown in Figure 2. The first step was to locate an intensity B-scan in an en face intensity image within a 10° wide circle (in diameter) centered on the ONH, which contained the largest blood vessel. The corresponding flow B-scan was also generated using color-Doppler analysis [33]. The second step was to locate the boundaries of the blood vessel based on intensity and flow data in B-scans and isolate a cross-sectional image of the vessel. After realigning the images with respect to the tissue surface (based on intensity thresholding [34]), intensity and retardation plots were generated by averaging A-lines within the isolated cross-sectional area. These plots were subsequently used to detect the inner edge of the blood vessel wall. Inside the blood vessel wall, the cumulative retardation increased linearly with depth and reached a plateau below the blood vessel wall edge. The intensity plot experienced a drop in intensity at this edge, helping to determine the thickness of the blood vessel wall. A drop in intensity is expected inside the blood vessel because of increased forward scattering in laminar blood flow, resulting in an attenuation of the signal [24]. The linear increase in the cumulative retardance inside the wall stems from its organized fibrous tissue and smooth muscle, while the blood flow induces no retardation beyond the wall–lumen interface, leading to a plateau in the DPPR [35]. To obtain DPPR/UD, the retardation ranging from the surface of the vessel wall to the edge of the wall were least-squares fit. Fitting ensured that our DPPR/UD values are less dependent on the thickness measurement [36,37]. In the next step, DPPR/UD was then converted to the dimensionless birefringence using
(1)$$\Delta n=\frac{\left(\frac{DPPR}{UD}\;\right)\lambda }{2\times 360˚},$$
where *λ* = 840 nm is the central wavelength of the source and a 2 factor to account for the double pass. *BBI* was finally obtained by dividing the blood vessel wall’s thickness by the birefringence squared:(2)$$BBI=\frac{thickness}{{\Delta n}^{2}}.$$

Accuracy and the precision of the instrument were between 0.72 to 0.82 µm and 0.87 to 1.22 µm, respectively, based on measurements obtained from a plane mirror, as reported in our previous study [24]. Furthermore, we also demonstrated that the analysis is reproducible and does not depend on the examiner; we demonstrated that there is high interobserver agreement between two examiners using Bland–Altman plots and intra-class correlation coefficient (ICC) [24,25]. In addition, in our previous study, we indicated that our thickness and DPPR/UD measurements are reliable [24].

As vessel wall thickness are likely to depend on vessel width, we took measurements to correlate the blood vessel wall thickness to the diameter of the blood vessel. Additionally, to analyze the impact of low or high SNR on the blood vessel wall birefringence measurements, SNR and birefringence were plotted for various situations.

### 2.4. Statistical Analysis

One-way analysis of variance (ANOVA) was performed with a custom-written script in MATLAB (2019b, Math Works, Natick, MA, USA) to calculate *p*-values. The 95th and 5th percentiles were also calculated in MATLAB and used to find the threshold to discriminate between healthy and diabetic blood vessel walls. The error bars throughout the paper indicate the standard deviation (SD). Pooled relative standard deviation (PRSD) and relative variations (PRV) used as measures of repeatability were also calculated based on a custom-written MATLAB script. No data points were excluded from the study.

## 3. Results

### 3.1. Thickness Behavior versus Blood Vessel Diameter

Blood vessel wall thickness (averaged over four consecutive B-scans) extracted from five healthy and five diabetic subjects were compared to the diameter of the corresponding blood vessels, to determine whether the blood vessel diameter affects the vessel wall thickness. The blood vessel wall thickness indeed increases linearly with vessel diameter (Figure 3a). This behavior was consistent for both arteries and veins and within all subjects, either diabetic or healthy.

To obtain a better understanding of the influence of this relationship on BBI values, we calculated $BBI^{\prime }$, which can be referred to as normalized BBI, based on a new variable which is calculated by dividing wall thickness and vessel diameter (BBI′=(thicknessdiameter)Δn2) and compared it to the BBIs calculated based on Equation (2) for the diabetic arteries, which in comparison to the other vessels represents the worst-case scenario (with the lowest R^2^ value). Figure 3b shows a linear relationship between $BBI^{\prime }$ and BBI, indicating that the width of a vessel does not affect BBI as a diabetes diagnostic biomarker. This suggests that any changes in one of the variables are expressed in the same way in the other variable, meaning that there are no inherent differences between BBI and $BBI^{\prime }$. These finding suggest that diameter of the blood vessels should not affect the thickness measurements (albeit within our selected windows: 10° wide circle around the ONH).

### 3.2. SNR and the Birefringence of the Blood Vessel Wall

Previous studies have shown that the accuracy of the birefringence measurements is highly dependent on the SNR [38,39]. To ensure reliable birefringence measurements, we investigated the impacts of SNR on the birefringence of the blood vessel walls in different healthy and diabetic subjects. Our results (Figure 4) show high SNR > 30 dB in all the blood vessels of healthy subjects. Moreover, at least five blood vessels from diabetic patients with SNRs > 30 dB had the same birefringence values of the diabetic blood vessels with SNRs < 30 dB. If we can assume constant birefringence in this cohort (our measurements indicate that birefringence is always higher than ~0.001 in diabetic vessel walls), an SNR of at least 25 dB provides reliable birefringence measurements. Empirically, a minimum SNR of 23 dB is required to extract a reliable birefringence value.

### 3.3. Thickness and DPPR/UD Comparison in Normal and Diabetic Blood Vessels

Realigned cross-sectional intensity and flow images with respect to the retinal surface of a diabetic patient and a healthy subject are shown in Figure 5a and Figure 2, respectively. The boundaries of the blood vessels were then located and isolated based on intensity and flow data (areas within the red lines). To determine the thickness and the DPPR/UD (birefringence) of the blood vessel walls, intensity and corresponding retardation images were first generated through averaging the A-scans inside the isolated area in Figure 5a and Figure 2. Next, the retardation curve (green curves in Figure 5b and Figure 2) was least-squares fit. The edge of this fit indicated the thickness of the blood vessel wall, while its slope represents the DPPR/UD. For the healthy subject the thickness and the DPPR/UD were 18 µm and 0.4°/µm, respectively resulting in a BBI of 73.5 m. These values for the diabetic patients were 15 µm, 1.8°/µm and 3.6 m, respectively. The difference in DPPR/UD and consequently BBI is quite large, and it originates from the difference in the slope of the fits in Figure 5b and Figure 2. Compared to the healthy subject that showed a fit with a moderate slope, in the diabetic patient, the slope is quite steep indicating a higher birefringence. Note that there does not seem to be much difference in the vessel wall thicknesses between the healthy and diabetic vessel walls.

### 3.4. BBI Measurements

BBI values for the different blood vessels superior and inferior to the ONH of recruited subjects (extracted based on (1)) are shown in Figure 6. BBI combines the thickness and birefringence values in one parameter as a measure for vessel wall tissue health. BBI for the diabetic patients were lower compared to the healthy subjects. The averaged BBI for the arteries extracted from the patients with diabetes was 10.1 ± 1.8 m. For comparison, averaged BBI for the arteries of normal subjects was 35.8 ± 4.5 m. For the veins, these values were 10.8 ± 2.1 m and 41.8 ± 3.6 m, respectively (Table 2). The difference in the BBIs stems from the fact that the blood vessel walls of diabetic patients are more birefringent, when compared to healthy vessel walls, while having a similar wall thickness (Table 2). According to (1), BBI is inversely proportional to the birefringence of the blood vessel walls, hence increased birefringence leads to larger BBIs. This may be related to the inflammation and fibrosis of the intima layer of the blood vessel walls in response to the elevated blood glucose.

As can be seen in Figure 6a,b, the BBI values of the retinal artery walls for the healthy subjects vary between 10.4 m and 65.4 m, while the patients’ BBI values were between 4.8 m and 16.0 m. Similar values obtained for the veins of healthy individuals were between 7.8 m and 67.5 m compared to the veins of the diabetic patients varying between 6.2 m and 15.1 m. These differences are all statistically significant with *p*-values < 0.001, presented in Figure 6a.

### 3.5. Classifying Thresholds Based on 95th and 5th Percentiles

The 5th and 95th percentiles were used to identify classifying thresholds based on the histograms represented in Figure 7. The 5th and 95th percentiles are values associated with the location within the dataset where 5% of the data are below or higher than that value, respectively. In our data, the lower threshold (5th percentile) was used to identify the healthy cohort and the higher threshold (95th percentile) was used to diagnose diabetes.

Based on the threshold values listed in Table 3, any artery with a BBI less than 15.6 m is diabetic. For veins, this value is 14.6 m. For an artery or vein to be identified as normal, the BBIs need to be higher than 15.8 m and 13.8 m, respectively. However, since there is an overlap in the vein threshold values (Figure 7b), the smaller threshold was chosen, meaning that BBIs larger than 13.8 are normal, where any smaller values are classified as diabetic.

### 3.6. BBI Values along a Vessel

To determine the variation of the BBI values along sections of two artery/vein pairs in one healthy and one diabetic subject a tracing procedure was adapted. As shown in Figure 8, the measured BBI values for the blood vessel walls of the normal individual (NM one and two) varied between 38 and 48 m, while the BBI of the diabetic patient’s blood vessel wall (DB one and two) ranged between 8 and 14 m. Furthermore, the variation along the vessel walls for vessels one to four in Figure 8 were 12%, 7%, 12% and 10%, respectively which were determined based on the ratio of the standard deviation over the averaged BBI.

### 3.7. Repeatability

Repeatability of the measurements was analyzed by extracting DPPR/UD, thickness and BBI values of a blood vessel that was imaged four times. Between recording each image, the subject was asked to relax and close their eyes for two minutes. The corresponding DPPR/UD, thickness and BBI results are shown in Figure 9. The PRSD and PRV were calculated for each measurement as a measure of the repeatability. For the DPPR/UD, these values were 3.6% and 0.12%, respectively, meaning that the variations are negligible. For the thickness measurements, the results showed 0% PRSD and PRV meaning that the thickness measurements are exceptionally repeatable. For the BBI, the values were 6% for both PRSD and PRV indicating acceptable repeatability. Furthermore, ANOVA analysis was performed to further evaluate the repeatability and *p*-values were calculated for each measurement. In ANOVA analysis, the null hypothesis was that the extracted DPPR/UD, thickness and BBI values for each image have the same mean value. Thickness measurements showed a *p*-value of ~1, whereas the DPPR/UD and BBI showed values of 0.88 and 0.86, respectively, confirming the null hypothesis.

### 3.8. Sensitivity, Specificity and Accuracy

Sensitivity in our method is defined as the probability of correctly identifying blood vessels of the diabetic patients. Likewise, specificity can be described as the probability of the method to correctly identify the blood vessels of normal individuals. Accuracy, on the other hand, is the ability of the method to classify the blood vessel in the correct groups. Sensitivity, specificity and accuracy of our method are defined as follows [40,41]:(3)$$Sensitivity=\frac{TP}{TP+FN}\;$$
(4)$$Specificity=\frac{TN}{TN+FP}\;$$
(5)$$Accuracy=\frac{TN+TP}{TP+TN+FP+FN}\;$$
where *TP* represents the blood vessels, which belong to diabetic patients and are correctly classified as diabetic (true positive), *FN* represents the blood vessels of patients with diabetes, which are classified as normal (false negative), *TN* represents the blood vessels, which are correctly classified as normal (true negative) and *FP* represents the blood vessels of normal subjects, which are classified in the diabetic group (false positive). The sensitivity and specificity when distinguishing between arteries of normal subjects and those from diabetic patients were 93% and 95%, respectively. For the veins, the method showed 93% sensitivity and 91% specificity. The accuracy of our proposed method for the arteries and veins were 94% and 92%, respectively.

## 4. Discussion

Changes in the caliber of the retinal blood vessels are an important diagnostic biomarker to investigate the impacts of different pathogenesis caused by systemic and nonsystemic abnormalities. Parameters such as the wall-to-lumen ratio and wall thickness have been previously used to diagnose systemic hypertension, diabetes and CVD-associated end-of-organ risks [42,43,44,45]. However, the structural organization of the retinal blood vessel walls affected by diabetes has not been investigated before, in vivo. In this study, we assessed the largest blood vessels near the ONH for possible structural alteration through differences in polarization between patients with diabetes and a healthy control group. Using PS-OCT imaging, we showed that the retinal blood vessel walls of diabetes patients are more birefringent compared to those of a healthy control group. The wall thickness, on the other hand, does not experience any statistically significant changes, in neither group. BBI combines birefringence (equivalent to the structural integrity) and thickness (a widely used biomarker) of the blood vessel walls into one single quantity, and it was lower for the blood vessel walls of diabetic patients.

The increase in the birefringence of the retinal blood vessel walls (and accordingly, a reduction in BBI) can be attributed to atherosclerosis. Diabetes significantly speeds up atherosclerosis by promoting inflammation and reducing blood flow [46,47]. In diabetic patients, a combination of high triglycerides, low high-density lipoprotein (HDL) and modified low-density lipoprotein (LDL) retained in the arterial intima recruits monocyte-derived macrophages (a component of innate immunity) that are then partially differentiated into lipid-laden foam cells. The remaining macrophages accumulate in the plaque [3,46]. Atherosclerosis, or the hardening of the arteries, was formerly thought to be caused by an excess of cholesterol that formed plaques in the arteries. Today, studies demonstrate that most of the problems originate from a response of the body’s immune system toward the fatty build-up rather than the build-up itself. As a result of immune cells attacking these fatty deposits, inflammation occurs, increasing the risk of plaque expansion, rupture, and blood flow obstruction [48,49]. Fibrosis, the excess accumulation of extracellular matrix in blood vessel wall tissue may be another reason for higher birefringence in diabetic patients. Hyperglycemia and insulin resistance can directly trigger synthesis of fibroblasts. They can also indirectly convert endothelial cells into fibroblast-like phenotypes [50]. Since our results were derived through empirical experiments, they cannot provide an adequate explanation for the observed increase in birefringence within the blood vessel wall tissue in diabetic patients. Therefore, further ex vivo or animal studies are required to verify the exact contributing factors and the pathological mechanism behind this birefringence increase.

Due to the accumulation of plaque and build-up, an average increase in birefringence of 94% occurred in the artery walls of the diabetic patients compared to the healthy subjects. Birefringence in the vein walls of the patients was also twice as high as the birefringence measured in the walls of the healthy control group (100%). However, the thickness of the artery and vein walls for both healthy and diabetic groups were almost equal.

Our thickness findings for the healthy control cohort and diabetic patients do not agree with previous studies, bolstering the importance of the birefringence and BBI measurements [24,51,52]. In fact, there is a discrepancy in the vessel wall thickness measurements in different studies. For instance, Chui et al. [53] using AOSLO measurements claimed that retinal arteries are more than 10 μm thicker than vein walls. In another study, Rim et al. [54] reported arteries and veins have walls with a thickness of 23.9 μm and 20.7 μm, respectively. Muraoka et al., on the other hand, claimed that the wall thickness for arteries and veins are 14.0 μm and 11.7 μm, respectively, [51]. Moreover, another study showed that retinal arteries and veins have walls ranged between 14.2 μm and 12.2 μm, respectively [52]. Also, using adaptive-optics (AO) imaging methods, Burns et al. [55] and Żmijewska et al. [45] concluded that diabetes causes an increase in the vessel wall thickness of 25% and 20%, respectively. In all cited studies, only a single imaging modality was applied to find the vessel wall thickness. Moreover, these modalities were all based on measuring reflected intensity. From our own experience with an imaging system that does not use adaptive optics for high-resolution imaging and is affected by speckle noise [25], accurate thickness measurements are not possible without access to the extra contrast provided by the retardation and birefringence data (which can be considered as an extension), which may be one reason behind the disagreement between the various OCT imaging studies. Furthermore, the mentioned studies that used AO imaging measured the side wall thickness as opposed to the top walls in our study. It is possible that vessels and vessel walls experience a heterogeneous thickness increase due to diabetes [56]. Moreover, the recruited patient in those studies were at later stages of diabetes as all of them were diagnosed with diabetic retinopathy. The patients in our study did not have diabetic retinopathy. Another difference is that the thinner blood vessels around the macula were analyzed in those studies, while we assessed the thicker blood vessels adjacent to the ONH. Our data in Figure 3 indicate that thinner blood vessels have thinner vessel walls.

Different studies have used retinal vasculature properties (retinal hemodynamics) as an indirect biomarker for diabetic retinopathy [57], type I diabetes [58] and diabetic macular edema [59]. These studies mostly used fundus photography [57], Doppler flowmetry [59], pulse wave velocity [60] and retinal vascular fractal dimensions. However, none of these methods looked at the pathological and structural alteration in the retinal blood vessel walls as a potential biomarker for diabetes.

To our knowledge, there are no studies on reporting birefringence and retardation as a measure of the blood vessel wall integrity for diagnosis of diabetes. In our previous study, we showed that for the healthy subjects, the birefringence of the artery and vein walls are approximately 6 × 10^−4^ and 7 × 10^−4^, respectively [24], which is consistent with the findings in the current study. The birefringence of artery and vein walls in our diabetic patient cohort were 12 × 10^−4^ and 12 × 10^−4^, respectively, which is an increase of approximately 100%.

We introduced BBI and used it as a single parameter that can quantitatively distinguish blood vessels of healthy people from patients with diabetes. For the arteries, averaged BBI for the diabetic patients was 10.1 ± 1.8 m, considerably lower than BBI for the healthy control group with average BBI of 35.8 ± 4.5 m (~250% reduction). Those average values for the veins were 10.8 ± 2.1 m and 41.8 ± 3.6 m, respectively, an even more significant decrease of about 300%. The difference between the BBI of vessel walls of healthy and diabetic patients was statistically significant (*p* < 0.001).

It is interesting to note that the increase in slope in the DPPR plots is observed throughout the full width of the vessel wall (Figure 5b). This seems to indicate that every layer in the vessel wall is similarly affected by diabetes, meaning that fibrosis occurs at the same magnitude independent of depth.

In this study, a region of interest (ROI) around the ONH (within a circle with a radius of 5° centered at ONH) was chosen to assess the blood vessels. This ROI area was chosen to fulfil two purposes: first, to keep the variation in the BBI to a minimum to ensure reliability and second, to maintain a consistent location in different subjects for assessing the blood vessels, since the diameter and wall thickness of blood vessels vary by moving further away from the ONH [61]. In Figure 8, we investigated the extent of variation in BBI along four different blood vessels adjacent to ONH of one healthy and one diabetic subject and showed that the BBI variation was less than 12%, which we considered to be acceptable since it was within the averaged standard deviation.

### 4.1. Strengths

The extra contrast that is provided by the polarization measurement provides information about the structural integrity of the vessel walls, and this information cannot be obtained with traditional intensity-based imaging methods. Therefore, the strengths of our method are the high sensitivity, specificity, accuracy and good repeatability of the BBI measurements, rendering BBI a robust quantity in discriminating between blood vessels of diabetic and healthy people for the diagnosis of diabetes. Sensitivity and specificity of 93% (on average) ensure the feasibility of the BBI measurements as a classifying tool that can be used in clinics to diagnose diabetes. In contrast to traditional blood glucose measurements for the diagnosis of diabetes (not the management of diabetes), the technology is completely non-invasive. Imaging takes just a few seconds and an autonomous version of the technology for the detection of the location of blood vessels to determine its birefringence and BBI could provide an immediate result without lab work, crucial for low-cost screening. Instead of determining the glucose level, our method provides a BBI value that is a measure of the vessel wall health to provide insight into the health of the patient.

### 4.2. Weaknesses

Note that for each eye, at least four blood vessels were analyzed and that often both eyes were imaged. Optical properties of the investigated blood vessels between diabetic and healthy subjects were statistically well separated to limit the number of patients to five subjects in diabetic cohort and ten in healthy category. The number of subjects recruited in our study closely matches those from one of the most recent glaucoma and diabetic retinopathy clinical studies [62]. Indeed, there are some studies performed with PS-OCT that include more subjects, but these studies only performed experiments on patient eyes [63,64]. However, a next study should include more patients with stages ranging from virgin diabetics or diabetic suspects to end-stage diabetics to understand the distribution of BBI at various stages of the disease.

Another perceived weakness is a lack of characterization of the subjects. The lack of detailed characterization of the diabetic subjects such changes in HbA1c levels, fasting and postprandial blood glucose levels, insulin sensitivity, insulin resistance, beta-cell function, and diabetic complications such as retinopathy, neuropathy, and nephropathy. However, all recruited subjects diagnosed with diabetes had a confirmed long-term diagnosis by a qualified physician and were undergoing medication to regulate their blood glucose. Moreover, the blood glucose of the healthy subjects was not assessed in the study.

We used the 5th and 95th percentiles to find the classifying thresholds. Although no overlaps are seen between the two groups based on the histograms in Figure 7, we opted for a more conservative approach that identifies classifying thresholds for each group (normal and diabetic) with a margin of error between the two. This approach was mainly considered due to the relatively small number of participants. One could argue that receiver operating characteristic (ROC) classifier curves would be a better option to identify thresholds for discriminating blood vessels. However, these curves require a larger number of data points [65] and consequently are not appropriate for the current study.

Another weakness of the current study may be the fact that the data analysis was not performed blindly. We demonstrated in our previous publication that the analysis does not depend on the examiner. We used Bland–Altman plots and ICC and showed there is high interobserver agreement between two examiners [24]. Moreover, as DPPR data were automatically least-squares fit, the operator can only affect the DPPR/UD measurements by intentionally estimating healthy blood vessels to be too thick (if too thin the DPPR/UD measurement is not affected), but our measurements show that the healthy vessel walls are not thicker than the diabetic vessel walls. This proves that the operator cannot affect the measurements.

Our healthy subjects were considered healthy as they were not diagnosed with any diseases prior to imaging and were not on any medications. However, it is still possible that some of the healthy subjects had underlying complications with a possibility of affecting the blood vessel wall polarization properties. This might be the reason behind the fact that some of the blood vessels from the healthy cohort were classified as diabetic (FP values).

As we mentioned earlier, the diabetic subjects recruited for the current study were not diagnosed with hypertension. As we already know, patients with hypertension have relatively higher BBI values compared to healthy people [25]. This makes BBI an effective indicator for health in a single four-second measurement, as diabetic vessels have lower BBI values than healthy subjects, while hypertensive patients have higher BBI values than healthy subjects.

Here, one can imagine that since diabetes causes high vessel-wall birefringence, a diabetic patient shifting towards hypertension could have lower birefringence values over time. These could at some point balance to normal birefringence values. In such a case, based on what we have seen so far, it would be logical that these patients have a thicker vessel wall than normal subjects and normotensive diabetic patients. Further studies can help to quantify the different contributions to thickness and birefringence in diabetic patients with hypertension.

## 5. Conclusions

We proposed a non-invasive, quick and relatively cheap method based on PS-OCT to assess the integrity of the retinal blood vessel walls to diagnose diabetes through the eye. We implemented BBI, a single quantitative number that combines the wall tissue birefringence (a measure reflecting tissue organization) and thickness (a well-known biomarker) to evaluate the impact of high blood glucose on the blood vessels. Our results indicated that BBI can be employed as classifying thresholds to quantitatively distinguish diabetic blood vessels from healthy vessels with extraordinary sensitivity and specificity.

These are the first in vivo measurements of the birefringence of diabetic blood vessel walls, and there is a considerable increase in vessel wall birefringence in comparison to healthy subjects. In our small group of subjects, the increase is uniform over the vessel wall, suggesting that the full thickness of the vessel wall is affected by diabetes.

Unlike non-invasive PS-OCT measurement that is inherent to the optical and biomechanical properties of the blood vessel wall tissue, blood sugar measurements are invasive, inaccurate and are easily affected by different factors such as the patient’s dietary habits or hematocrit variations.

OCT is offered by some optometrists as a service, and something similar could be envisioned for a PS-OCT instrument for blood vessel wall measurements, providing a cheap screening tool for the patient’s health, to diagnose diabetes and hypertension.

## 6. Patents

BC Massachusetts General Hospital (P). BC and HA Yonsei University (P).

## Figures and Tables

**Figure 1 biomolecules-13-01230-f001:**
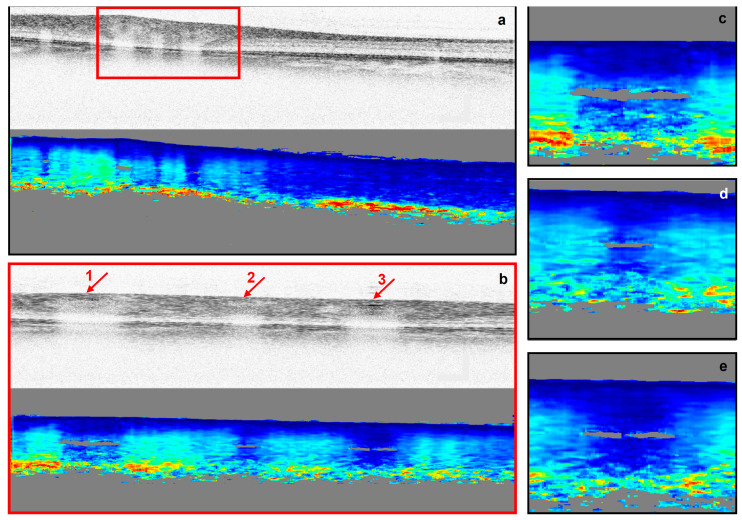
Intensity (gray-scaled) and retardation cross-sectional images of the right eye of a diabetic subject. (**a**,**b**) are the cross-sectional images of an area near the ONH imaged with different scanning sizes (**a**) 4.5 mm by 1.5 mm, (**b**) ~1.0 mm by 1.5 mm). (**c**–**e**) are magnified regions of blood vessels 1 to 3 indicated in (**b**). In intensity images, darker areas represent a higher intensity. In retardation images, blue to red represent lower (0°) to higher (180°) retardation.

**Figure 2 biomolecules-13-01230-f002:**
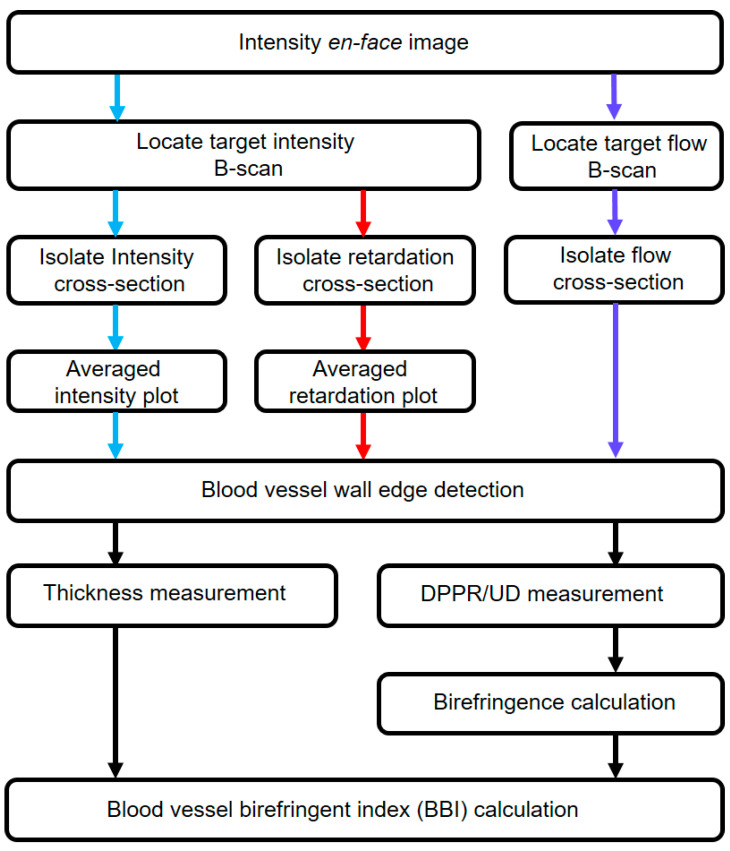
Diagram showing different steps taken to extract blood vessel wall polarization properties and corresponding BBI. Blue, red and purple arrows indicate steps taken based on the intensity, retardation and flow images, respectively. Black arrows show analysis based on blood vessel edge detection.

**Figure 3 biomolecules-13-01230-f003:**
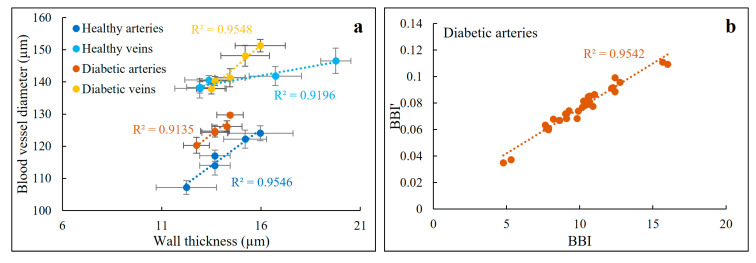
(**a**) Blood vessel diameters of five different blood vessels recorded from five healthy and five diabetic subjects plotted against corresponding wall thickness values. These data were averaged over four adjacent B-scans. There is a linear relationship between blood vessel diameter and corresponding wall thickness within all groups with R^2^ values ranging from 0.91 to 0.95. Error bars show SD. (**b**) Variations of BBI’ (i.e., normalized BBI) versus BBI follows a linear relationship with an R^2^ value of 0.95.

**Figure 4 biomolecules-13-01230-f004:**
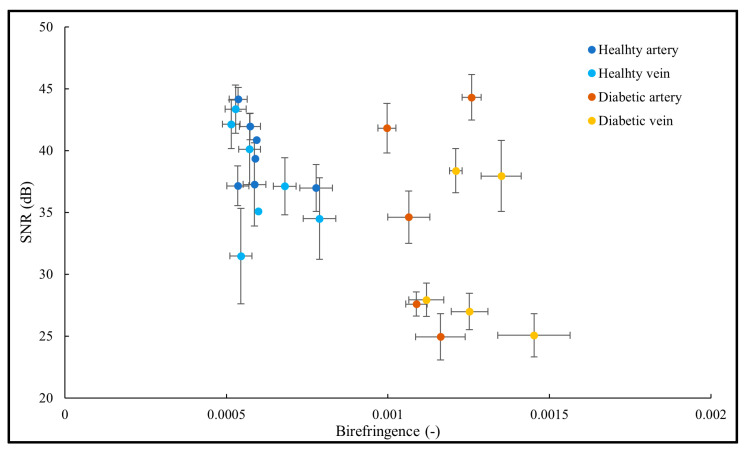
Birefringence of the blood vessel wall plotted against SNR of the area containing the blood vessel wall. While all the healthy blood vessels have higher SNRs, the birefringence values of the diabetic blood vessels are consistent and do not vary as a function of SNR. SNR and birefringence values are averaged over four adjacent B-scans. Error bars indicate SD.

**Figure 5 biomolecules-13-01230-f005:**
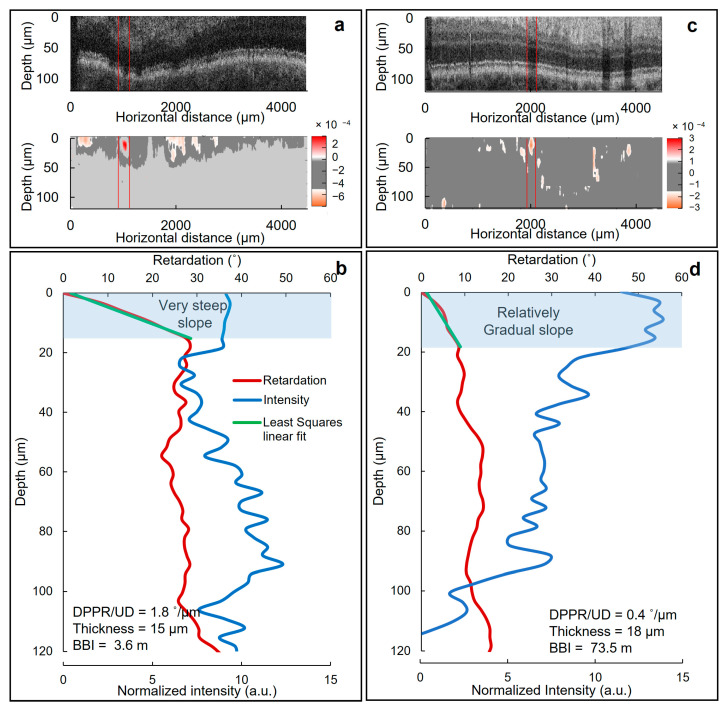
Thickness and DPPR/UD measurements of diabetic and normal vessel walls. Realigned intensity (top panel) and corresponding color-Doppler flow (bottom panel) cross-sectional images with respect to the retinal surface recorded from (**a**) a diabetic patient and (**c**) a healthy individual. In the intensity images, a darker color indicates a lower intensity. The areas between two red lines mark the isolated blood vessels for analysis. (**b**) Associated DPPR and intensity images obtained by averaging A-lines within the isolated blood vessel area in (**a**). (**d**) Associated DPPR and intensity images generated by averaging A-lines within the isolated blood vessel area in (**c**). The DPPR is the product of DPPR/UD (birefringence) and optical path length. A least-squares fit was used to calculate DPPR/UD and to determine the boundary and therefore thickness of the blood vessel wall. In (**b**), the fit has a steep slope meaning that the diabetic blood vessel wall induced higher birefringence compared to the relatively moderate slope of the healthy blood vessel wall in (**d**). Blue-highlighted areas indicate blood vessel wall tissue.

**Figure 6 biomolecules-13-01230-f006:**
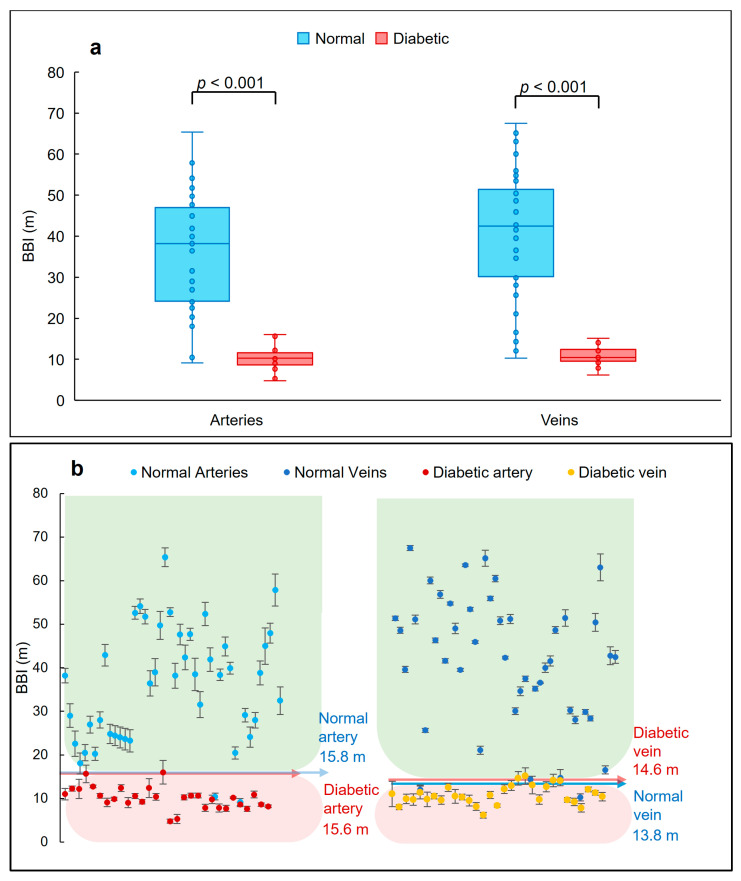
BBI data obtained from normal and diabetic artery and vein walls. (**a**) Distribution of the BBI in healthy and diabetic subjects in different retinal arteries and veins. In diabetic patients, the BBI experienced a reduction in comparison to normal tissue. This reduction was higher in veins. The boxplots were generated based on the median, lower and upper quartiles. Bulletpoints show the distribution of the data and whiskers indicate the minimum and maximum values. No outlier points were identified in the plots. The difference between normal and diabetic BBIs are also statistically significant (*p* < 0.001). (**b**) Plots showing the BBI values of arteries and veins extracted from normal and diabetic subjects. Data are represented as mean ± SD (averaged over four adjacent B-scans). Arrows show the thresholds obtained by determining the 5th and 95th percentiles. Green and red backgrounds mark the normal and diabetic values. Statistical analyses were performed using one-way ANOVA.

**Figure 7 biomolecules-13-01230-f007:**
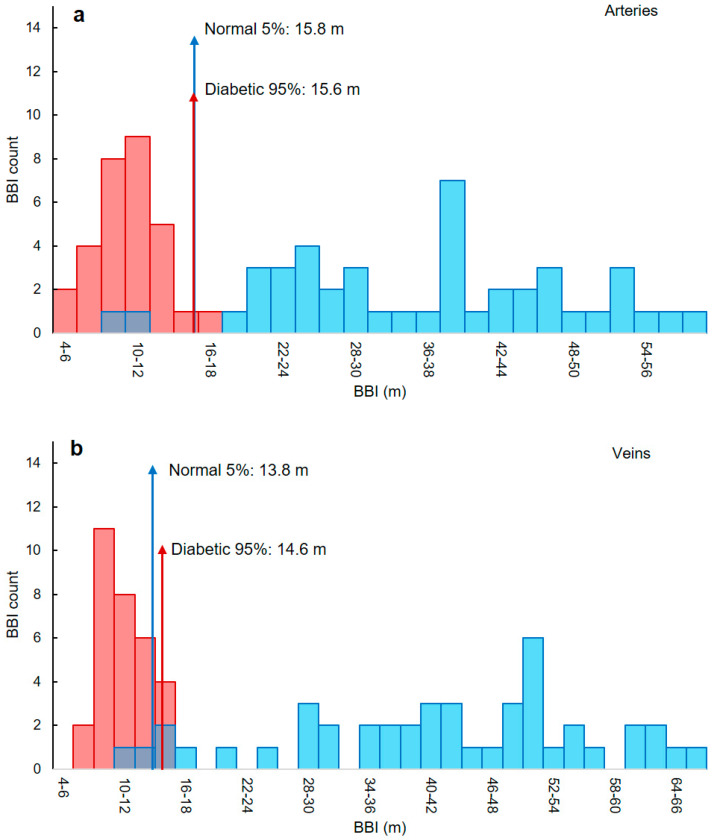
Thresholds used to discriminate healthy and diabetic subjects for artery and vein walls. Histogram representation of the extracted BBI values for the analyzed (**a**) arteries and (**b**) veins of healthy (blue) and diabetic (red) subjects. The 5th and 95th percentiles were used to find the thresholds to distinguish between blood vessels of healthy subjects and from patients with diabetes (red and blue arrows). For arteries, BBI values smaller than 15.6 m are considered diabetic, while BBIs higher than 15.8 m are normal. For veins, the threshold is at 13.8 m.

**Figure 8 biomolecules-13-01230-f008:**
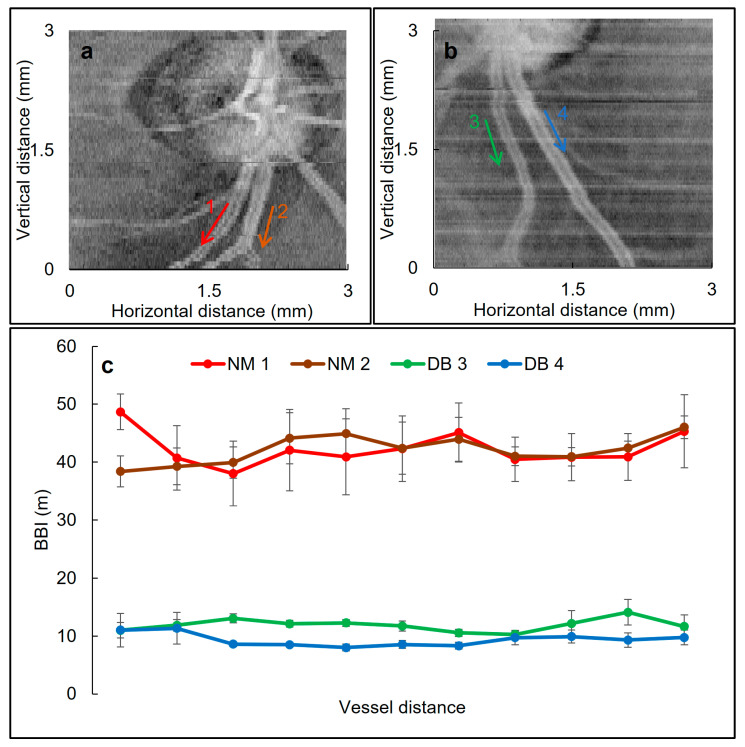
Variation of the BBI values along different blood vessels. Cropped intensity en face images of the right eyes of (**a**) a healthy subject and (**b**) a diabetic patient. The arrows indicate the blood vessels where BBI was obtained. (**c**) BBI variation along the indicated blood vessels. Each value is the average of measurements obtained from four adjacent B-scans. Error bars show SD. NM: normal, DB: diabetic.

**Figure 9 biomolecules-13-01230-f009:**
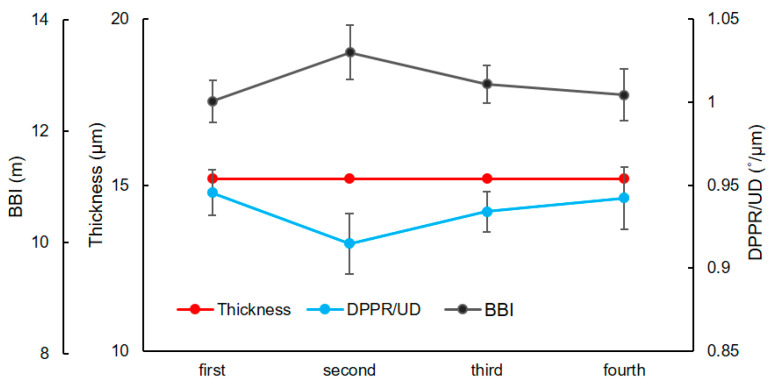
Repeatability of the thickness, DPPR/UD and BBI measurements. A blood vessel from the right eye of a diabetic subject was imaged four times and used to evaluate repeatability. Error bars are the SD of four adjacent B-scans. Based on ANOVA analysis, PRV for DPPR/UD, thickness and BBI were 0.12%, 0% and 6.3%, with *p*-values of 0.88, ~1 and 0.86, respectively, meaning that the measurements are repeatable.

**Table 1 biomolecules-13-01230-t001:** Characteristics of the study subjects (N = 15).

Characteristics	Normal (N = 10)	Diabetes (N = 5)
Age ± SD, y	49 ± 11	47 ± 8
Age range, y	30 to 64	33 to 59
Eyes, N	18	10
Sum of vessels analyzed (artery/vein)	44/45	30/31
Sex (female/male), N	4/6	2/3
Height, mean ± SD, cm	178 ± 8	173 ± 9
Body mass index ± SD, kg/m^2^	25.3 ± 2.1	27.3 ± 2.2
Smoking status		
Never, N (%)	7 (70%)	4 (80%)
Former, N (%)	2 (20%)	1 (20%)
Current, N (%)	1 (10%)	0 (0%)
Clinical characteristics		
Type I/II	N/A	0/5
Controlled/uncontrolled	N/A	5/0
Disease duration ± SD, y	N/A	3.5 ± 1.3
Disease duration, range, y	N/A	1.5 to 7
Diabetic retinopathy	N/A	0
Hypertension	N/A	0

**Table 2 biomolecules-13-01230-t002:** Average BBI values extracted from healthy people and patients with diabetes. Retinal blood vessels of diabetic patients showed lower BBIs compared to vessel walls of healthy subjects.

	BBI Average Values (m)			Average Birefringence (×10^−4^)			Average Thickness (µm)		
	Artery	Vein	All	Artery	Vein	All	Artery	Vein	All
Diabetic ± SD	10.1 ± 1.8	10.8 ± 2.1	10.5 ± 1.9	12.4 ± 1.7	11.8 ± 1.1	12.1 ± 1.4	14 ± 1	15 ± 1	15 ± 1
Normal ± SD	35.8 ± 4.5	41.8 ± 3.6	38.8 ± 4.1	6.4 ± 1.2	5.9 ± 0.8	6.2 ± 1.0	15 ± 2	15 ± 2	15 ± 2
*p*-value	<0.0001	<0.0001	<0.0001	<0.0001	<0.0001	<0.0001	0.68	<0.05	<0.05

**Table 3 biomolecules-13-01230-t003:** Classification of healthy and diabetic subjects based on BBI thresholds. Thresholds were calculated based on the 5th and 95th percentiles shown in Figure 7. Any BBI smaller than the threshold in the red boxes is considered diabetic; likewise, any value larger than the threshold indicated in the green area is normal.

BBI Thresholds (m)		
	Diabetic	Normal
Artery	<15.6	>15.8
Vein	<13.8	>13.8

## Data Availability

The data presented in this study are available on request from the corresponding author.

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
