# Peer review of "Non-Invasive Retinal Blood Vessel Wall Measurements with Polarization-Sensitive Optical Coherence Tomography for Diabetes Assessment: A Quantitative Study"

_biomolecules, 2023, doi:10.3390/biom13081230_

Round 1

Reviewer 1 Report

The authors have well written the manuscript titled "Non-invasive retinal blood vessel walls measurements with polarization-sensitive optical coherence tomography for diabetes assessment: a quantitative study" 

here are a few comments:

1. Introduction about the rationale for measuring vessel parameters would be good to add.

2. in methods, page 3, line 103: no reference found?

3. why the study sample is very small given that the availability of normal and diabetic eye are much high?

4. authors could have increased the number of eyes to draw reliable conclusions not affected by the lower sample 

5. what is the variability of the study vessel parameters within the same eye in normals?

6. statistics section: why the authors used a parametric test with such a small sample? who not a non-parametric test to compare?

7. Results: how many blood vessels in the eyes were included in the analysis?

8. Better to have p values in the comparison tables.

Author Response

Comment 1. 1. Introduction about the rationale for measuring vessel parameters would be good to add.

Response:

The following text was added to the introduction section of the manuscript (page 1, line 45):

Evaluating the structural and functional characteristics of blood vessels becomes crucial for early detection of vascular disorders [11]. Recent investigations have revealed that alterations in blood vessel architecture can precede the onset of CVDs [12, 13]. The loss of blood vessel wall elasticity may arise due to aging, calcification, changes in collagen fibers and elastin composition, or inflammation [14, 15].

Also, the abstract was also modified. The new abstract (page 1, line 25):

 “Abstract: Diabetes affects the structure of the blood vessel walls. Since the blood vessel walls are made of birefringent organized tissue, any change or damage to this organization can be evaluated using polarization sensitive optical coherence tomography (PS-OCT). In this paper, we used PS-OCT along with the blood vessel wall birefringence index (BBI = thickness/birefringence2) to non-invasively assess the structural integrity of the human retinal blood vessel walls in patients with diabetes and compared the results to those of healthy subjects. PS-OCT measurements revealed that blood vessel walls of diabetic patients exhibit a much higher birefringence while having the same wall thickness and therefore lower BBI values. Applying BBI to diagnose diabetes demonstrated high accuracy (93%), sensitivity (93%) and specificity (93%). PS-OCT measurements can quantify small changes in the polarization properties of retinal vessel walls associated with diabetes, which provides researchers with a new imaging tool to determine the effects of exercise, medication, and alternative diets on the development of diabetes.

Comment 1. 2. in methods, page 3, line 103: no reference found?

Response:

The “no reference found” is referring to Table 1. This was fixed in the new version.

Comment 1. 3. why the study sample is very small given that the availability of normal and diabetic eye are much high?

Response:

We acknowledged the limited sample size as a weakness in the discussion of the manuscript. While Covid-19 played a role, this is a preliminary study to assess feasibility.  Nobody else has ever looked at diabetic vessel walls with PS-OCT before, so to obtain sufficient statistical power we had to know the difference in vessel wall thickness and birefringence between the two groups. Studies on a large group of subjects are very expensive. Conducting research involving a larger sample size at this time, considering the manual analysis of the PS-OCT images and manual procedure to extract the polarization properties of the blood vessels, was not possible. Note that we had 5 diabetic patients and 10 controls. In most of our published studies, we have tested the feasibility of measuring retinal tissue properties on fewer subjects, and these studies are well cited:

  • Retinal nerve fiber layer (3 normal subjects), 418 citations [1]
  • Henle fiber layer (3 normal subjects), 38 citations [2]
  • Retinal pigment epithelium (10 older subjects, 10 younger subjects), 14 citations [3]
  • Hypertension (10 hypertensives, 10 controls) [4]

Most of these studies have recently been reproduced by our competitors from the University of Vienna, who performed similar studies on larger samples, with the same results:

  • Henle fiber layer (150 healthy controls) [5]
  • Retinal nerve fiber layer (50 healthy controls) [6]
  • Retinal pigment epithelium, 153 healthy controls (2022) [6]

While the number of patients in the current study was small, the magnitude of the difference in vessel wall birefringence between healthy subjects and diabetics was almost 100% (6.2 ± 1.0 vs. 12.1 ± 1.4 with p-value < 0.0001) with a high consistency in each group.

In our future research, we plan to expand the sample size and include a more diverse group of participants to better represent the broader population. By doing so, we aim to strengthen the external validity of our findings and provide more conclusive results that can be applied more broadly.

To address the concern, the following paragraphs were added to the discussion part (page 16, line 495):

Our study was performed on only five patients with diabetes. Note that for each eye, at least 4 blood vessels were analyzed and that often both eyes were imaged. Optical properties of the investigated blood vessels between diabetic and healthy subjects were statistically well separated to limit the number of patients to five subjects in diabetic cohort and ten in healthy category. The number of subjects recruited in our study closely matches those from one of the most recent glaucoma and diabetic retinopathy clinical studies [61]. Indeed, there are some studies performed with PS-OCT that include more subjects, but these studies only performed experiments on patient eyes [62, 63]. However, a next study should include more patients with stages ranging from virgin diabetics or diabetic suspects to end-stage diabetics, to understand the distribution of BBI at various stages of the disease.

Another perceived weakness is a lack of characterization of the subjects. The lack of detailed characterization of the diabetic subjects such changes in HbA1c levels, fasting and postprandial blood glucose levels, insulin sensitivity, insulin resistance, beta-cell function, and diabetic complications such as retinopathy and neuropathy, nephropathy. However, all recruited subjects diagnosed with diabetes had a confirmed long-term diagnosis by a qualified physician and were undergoing medication to regulate their blood glucose. Moreover, the blood glucose of the healthy subjects was not assessed in the study.”

Comment 1. 4. authors could have increased the number of eyes to draw reliable conclusions not affected by the lower sample 

Response:

Please refer to our response to comment 1.3.

Comment 1. 5. what is the variability of the study vessel parameters within the same eye in normals?

Response:

We showed the variability within one image in Figure 8a, where two blood vessels of a healthy subject are evaluated. According to the results, the variation along the vessel walls were 12 and 7 percent, respectively, for blood vessel number 1 and 2. Also, in Figure 9, we showed the repeatability of the method in one diabetic subject.

In addition, to address the reviewer concern, we performed an intra-sex evaluation of BBI for healthy subjects and it shows no difference in the reported BBI values (Figure below).

“Figure 1. Intra-sex variation of BBI in the healthy cohort. BBI data were extracted from blood vessel walls of four male and six female subjects. Green and blue bars represent BBI for female and male subjects, respectively, and red lines represent the averaged BBI value per sex. A t-test analysis was conducted, assuming the null hypothesis that there is no significant difference between the means of BBI for males and females. For both arteries and veins, BBI is independent of sex with p-values of 0.92 and 0.21. With p-values greater than 0.05, the null hypothesis was not rejected, meaning that there is no significant difference between the means of BBI for males and females, and thus, BBI is independent of sex.

Comment 1. 6. statistics section: why the authors used a parametric test with such a small sample? who not a non-parametric test to compare?

Response:

Thanks for your comment. The authors acknowledge the use of parametric statistical tests as superior to other non-parametric tests for the following reasons:

  1. Our data distribution is normal and has a homogeneous variance. In this case parametric tests (e.g. ANOVA or t-test) are more robust than non-parametric ones such as Wilcoxon rank-sum test or the Kruskal-Wallis test.
  2. Non-parametric tests are in general less powerful than their parametric counterparts. Since our sample size is small and meets parametric tests assumption, we chose a parametric test to increase sensitivity and improve the chances of detecting significant effects.
  3. Parametric tests are more commonly used and are well-established facilitating easy understanding.

Comment 1. 7. Results: how many blood vessels in the eyes were included in the analysis?

Response:

For each eye at least four blood vessels were included in the studies. The number of the eyes included in the study is listed in Table 1. In total, 18 healthy eyes (89 blood vessels – 44 arteries and 45 veins) and 10 diabetic eyes (61 blood vessels – 30 arteries and 31 veins) were included in the analysis.

Comment 1. 8. Better to have p values in the comparison tables.

Response:

Thanks for your comment, p-values were added to Table 2. This table was modified to accommodate the change. The new table is shown below (page 11, line 297):

BBI average values (m)

Average birefringence (× 10-4)

Average thickness (µm)

Artery

Vein

All

Artery

Vein

All

Artery

Vein

All

Diabetic ± SD

10.1 ± 1.8

10.8 ± 2.1

10.5 ± 1.9

12.4 ± 1 .7

11.8 ± 1.1

12.1 ± 1.4

14 ± 1

15 ± 1

15 ± 1

Normal ± SD

35.8 ± 4.5

41.8 ± 3.6

38.8 ± 4.1

6.4 ± 1.2

5.9 ± 0.8

6.2 ± 1.0

15 ± 2

15 ± 2

15 ± 2

p-value

<0.0001

<0.0001

<0.0001

<0.0001

<0.0001

<0.0001

0.68

<0.05

<0.05

Reviewer 2 Report

This is a very interesting article concerning a non-invasive retinal blood vessel walls measurements with polarization sensitive optical coherence tomography for diabetes assessment, compared to healthy subjects.

Even if the topic could be considered very challenging and interesting, as well as its originality and novelty, the article is based only on the study of 5 diabetic patients, on whom a statistical analysis was also performed. Five patients are really too few to draw any conclusions (even if the authors themselves cited this as a limitation of the study).

In my opinion, this paper should be published more as a case series rather than a scientific article, with a different text structure.

For this reason, although this paper could be considered really oroginal, it can not be accepted as scientific article but it should be published in another article type (case series or communication), unless the study sample is increased.

Minor editing of English language required

Reviewer 3 Report

1)       The criterion for subjects of type II diabetes and healthy control was not rigorous. Also, the sample size was small. All the information mentioned above may influence the comparison result.

2)       Diabetic retinopathy severity may also influence the comparison result.

3)       Line 118: the author should provide more details about how to decide the threshold.

4)       What causes the horizontal distortions in Figure 8 (a) and (b)?

5)       How about the reliability of this method? The method's reliability will impact the diagnostic ability of this biomarker, especially for disease screening.

The author could improve the language writing level.

Round 2

Reviewer 2 Report

The paper is now suitable for publication.